# Isolation of Hepatocytes from Liver Tissue by a Novel, Semi-Automated Perfusion Technology

**DOI:** 10.3390/biomedicines10092198

**Published:** 2022-09-06

**Authors:** Carsten Poggel, Timo Adams, Ronald Janzen, Alexander Hofmann, Olaf Hardt, Elke Roeb, Sarah K. Schröder, Carmen G. Tag, Martin Roderfeld, Ralf Weiskirchen

**Affiliations:** 1Miltenyi Biotec B.V. & Co. KG, R&D Department, 51429 Bergisch Gladbach, Germany; 2Department of Gastroenterology, Justus-Liebig-University Giessen, 35392 Giessen, Germany; 3Institute of Molecular Pathobiochemistry, Experimental Gene Therapy and Clinical Chemistry (IFMPEGKC), RWTH University Hospital Aachen, 52074 Aachen, Germany

**Keywords:** primary liver cells, hepatocyte, automated isolation, cell separation, cell culture, perfusion

## Abstract

Primary hepatocytes are a major tool in biomedical research. However, obtaining high yields of variable hepatocytes is technically challenging. Most protocols rely on the two-step collagenase perfusion protocol introduced by Per Ottar Seglen in 1976. In this procedure, the liver is perfused in situ with a recirculating, constant volume of calcium-free buffer, which is maintained at 37 °C and continuously oxygenated. In a second step, the liver is removed from the carcass and perfused with a collagenase solution in order to dissociate the extracellular matrix of the liver and liberate individual cells. Finally, the dissected hepatocytes are further purified and concentrated by density-based centrifugation. However, failure in proper cannulation, incomplete enzymatic digestion or over-digestion can result in low cell yield and viability. Here we present a novel semi-automated perfusion device, which allows gentle, rapid and efficient generation of a single-cell suspension from rodent livers. In combination with prefabricated buffers, the system allows reliable and highly reproducible isolation of primary hepatocytes.

## 1. Introduction

Primary hepatocytes are a well-established in vitro model widely used to investigate numerous aspects of liver physiology and pathology. Cultured hepatocytes are still considered as the gold standard for many applications because they carry out most of the hepatic functions including protein synthesis, protein storage, carbohydrate metabolism and synthesis of cholesterol, bile salts and phospholipids [1,2]. They have become a standard model to evaluate hepatic drug metabolism and there is a promise that hepatocyte organoids derived from either fetal or adult livers will have great impact on all areas of regenerative medicine [3,4]. In this regard, hepatocyte transplantation with in vitro-expanded hepatocytes has been intensively explored to treat liver diseases.

During the last decade, various approaches have been employed to promote the viability and functionality of primary hepatocytes in culture. In particular, two-dimensional (2D) sandwich cultures, diverse co-culture systems with non-parenchymal liver cells, and aggregation into three-dimensional (3D) organoids have been established and have improved the ability to expand hepatocytes in culture for longer periods [4,5]. In addition, primary hepatocytes are an integral part of several “liver-on-a-chip” and “liver-disease-on-a-chip” models that have become valuable tools to study liver disease, facilitate drug discovery, and enable toxicity testing [6]. Nevertheless, all these encouraging methods are still in progress and further research aiming to improve techniques for isolation of viable primary hepatocytes is necessary.

The landmark study by Seglen has shown that the collagenase perfusion method is likely the ultimate method for preparation of hepatocytes [7]. In the respective procedure, the liver is first pre-perfused with a Ca^2+^-free buffer to wash out blood and circulating cells and further disperse the intracellular matrix of the liver tissue by loosening the cell-cell connections. Thereafter, the liver is perfused with a buffer containing Ca^2+^ and collagenase in order to dissociate the extracellular matrix and liberate individual cells from the liver tissue.

Although the method is straightforward, there are several potential pitfalls that might result in overall low cell yield and viability. First, the cannulation process is challenging, particular when working with young animals that have smaller veins. Second, batch-to-batch variation of collagenase might result in unpredictable enzymatic efficiencies and low reproducibility, requiring fine-tuning of individual collagenase lots for the standardized cell isolation process. Third, the isolation of primary cells using a perfusion apparatus and self-made buffers is prone to biological contamination.

Consequently, the isolation of viable primary hepatocytes is technically demanding and requires trained and qualified staff. Moreover, anesthetized animals subjected to in situ perfusion by cannulation of the portal vein are placed in the USDA pain/distress category D in the US as it is considered a painful procedure, and as a surgical model performed under general anesthesia, it is classified as a “moderate procedure” in the EU; in both cases, this requires review and approval by responsible animal welfare committees [8,9,10]. Nevertheless, the pioneering protocol of Seglen for isolation of rat hepatocytes is widely applied and has been adapted to mice and humans [11,12,13,14,15,16].

## 2. Results

To increase the reproducibility and to simplify the cell isolation procedure, we have developed a new disposable semi-automated perfusion tube (i.e., the gentleMACS^TM^ Perfuser) and Perfusion Sleeves for the gentleMACS Octo Dissociator with Heaters (Miltenyi Biotec, Bergisch Gladbach, Germany), which can be run with preformatted, standardized reagents to allow simple and reliable isolation of murine primary hepatocytes.

The system is composed out of four individual parts (i.e., Lid, Clamp, Grid, and Base). The liver tissue is clamped on the Grid via the Clamp, and the Lid is connected to the Grid by snap-fit before being fixed to the Base by thread-lock (Figure 1A–C).

The usage of the gentleMACS Perfusion Technology system is rather simple and semi-automated. In brief, the gentleMACS Perfuser comes with a Lid attached to a Base and a disassembled Grid and Clamp (Figure 2A). At the beginning of a perfusion, the gentleMACS Perfusion Sleeves are installed on the gentleMACS Octo Dissociator with Heaters, the gentleMACS Perfuser Base-Lid (w/o Clamp and Grid) is placed on one of the eight positions of the instrument and the gentleMACS Heating Unit is put on the Perfuser. Then, the liver lobe is placed onto the Grid and fixed with the Clamp (Figure 2B,C). Afterwards, the Lid containing the Grid holder is connected to the Grid-Tissue-Clamp assembly by simply clicking the two ends of the Grid holder into the Grid (Figure 2D). In the next step, the Lid-Grid-Tissue-Clamp assembly is transferred to the gentleMACS Perfuser Base. By screwing the Lid via the thread into the Base, the four needles of the Base automatically penetrate the liver tissue.

Following this, the gentleMACS Octo Dissociator with Heaters runs an automated program (37C_m_LIPK_1) optimized for digestion of mouse liver which consists of different steps including priming, initial perfusion, washing, equilibration, and enzymatic perfusion (Figure 2E,F). At different steps, the program pauses allowing the operator to manually add or remove buffers or enzyme solutions using a disposable glass Pasteur pipette with elongated tips (Figure 2G,H). After each step, the program is resumed via the display of the gentleMACS Octo Dissociator with Heaters (Figure 2I).

After completion of the perfusion process, the used enzyme solution and the perfused liver lobe are transferred to the gentleMACS C Tube, an established tube format for dissociation purposes, which can be run with standard sleeves on the gentleMACS Octo Dissociator with Heaters (Figure 2J–N). In this disposable tube, the hepatocytes are released from the perfused liver tissue by soft stirring, leading to turbidity of the solution (Figure 2O). Finally, the cell suspension is filtered through a MACS SmartStrainer and hepatocytes are enriched and collected by a low-spin centrifugation (Figure 2P,Q).

Using this procedure, the typical yield isolated from one liver lobe as estimated from 18 individual experiments was ~1.4 × 10^7^ hepatocytes with a ~83% viability as assessed by trypan blue exclusion staining.

Another round of independent experiments using the left lateral liver lobes of female mice (n = 33) with variable genetic backgrounds as starting materials resulted in 1.1 × 10^7^ ± 3.7 × 10^6^ hepatocytes (corresponding to 3.4 × 10^7^ ± 1.3 × 10^7^ hepatocytes per gram of liver tissue) with a purity of 88.96% ± 3.00% as assessed by flow cytometry (Table 1).

Similar to the conventional isolation procedure, the success of washing and enzymatic digestion can be followed by the typical color change of the liver tissue (Figure 3).

The isolated hepatocytes have the characteristic flattened, hexagonal appearance after 24 h following plating (Figure 4).

Immunofluorescence analysis and Western blot analysis revealed that hepatocytes isolated with the gentleMACS Perfusion Technology express typical parenchymal markers including the hepatocyte cytokeratin 8, Zonula occludens-1 required for tight junctions and apical polarity in hepatocytes and hepatocyte nuclear factor 4α acting as a master regulator of hepatic differentiation [17,18,19] (Figure 5).

Similar to hepatocytes that were isolated by the conventional perfusion protocol, cells isolated with the gentleMACS Perfusion Technology are capable of expressing mRNA for albumin (*Alb*) and hepatocyte nuclear factor 4α (*Hnf4a*) (Figure 6A). Moreover, the cells express and secrete Lipocalin 2 (LCN2), a protein that can be further triggered by the pro-inflammatory cytokine interleukin-1β (IL-1β) (Figure 6B)—a finding that was previously reported by us and others—and help to counteract IL-1β-induced stress [20,21]. In addition, the isolated cells express hepatic estrogen receptor α (ER-α) which plays a key role in the maintenance of gluconeogenesis and lipid metabolism [22]. Similar to cells isolated by the conventional isolation procedure, the cells can be grown for several days in-culture. There were also no differences between the hepatocytes isolated by the two different isolation methods in regard to sensitivity towards IL-1β (Figure 6C).

The phenotypic appearance and the marker expression indicate that the cells isolated with the gentleMACS Perfusion Technology are similar to those isolated by usage of a conventional perfusion apparatus.

## 3. Discussion

The automated procedure is easy, operator-independent, and does not require extensive training of personnel. The system allows parallelization of up to eight perfusions at the same time on one instrument. In addition, the disposable format of the Perfuser in combination with optimized reagents and enzyme solutions offers an easy aseptic workflow with strict lot-to-lot consistency. Finally, the Perfuser allows isolation of cells from a liver or parts thereof taken from dead animals (ideally within 5 min of sacrifice). This reduces the suffering of animals, and in many countries, there is no need to obtain permission for perfusion from any animal ethics committee. Importantly, remaining liver parts can be used further for other applications, reducing the overall number of animals used for scientific purposes. Therefore, the device is a significant add-on to foster the 3R principle proposed by Russell and Burch to avoid animal experiments (Replacement) and to limit the number of animals (Reduction) and their suffering (Refinement) [23].

We are convinced that the Perfuser, which allows the ex vivo perfusion of any tissue using enzymes, will also find other wide-ranging applications. In fact, we are working on protocols that will allow us to isolate viable hepatocytes from diseased liver tissue. Moreover, it is reasonable to assume that the device will allow the isolation of non-parenchymal hepatic cell subtypes including hepatic stellate cells, liver sinusoidal endothelial cells, and Kupffer cells from other species and human materials. Accurate adaptation of enzyme concentrations will certainly also allow isolation of cells from fibrotic or even cirrhotic tissue.

The automation of the workflow has several additional advantages compared to conventional isolation procedures. In biomedical research, the overall reproducibility of experiments is important and automated protocols might help to prevent human failures. In this regard, a survey of 1576 researchers who took a brief online questionnaire on reproducibility in research showed an alarming trend: more than 70% of researchers failed to reproduce another scientist’s research, and more than half failed to reproduce their own experiments [24]. In light of the fact that cell-based systems have increasingly replaced in vivo research and the application of in vitro models has gained an ever-growing popularity [25], it is essential that primary cells isolated in different laboratories should be as equal as possible. However, the individual steps for procuring primary cells are experimentally challenging. In particular, failures in the cannulation of the vena cava and low cell viability resulting from either incomplete or over-digestion with collagenase are major pitfalls that might occur during conventional cell isolation [12]. Importantly, the gentleMACS Perfusion Technology uses pre-formatted buffers, standardized enzyme solution, and precisely controlled incubation times for priming, initial perfusion, washing, equilibration and enzymatic perfusion. The method further offers the possibility of using young animals as starting material because there is no need for in situ cannulation. Instead, the liver is taken from sacrificed animals, which simplifies the ethical framework for animal experimentation because in most countries animal ethics approval is not necessary for work with materials from dead animals.

Noteworthy is the fact that different lengths of tubing and flow rates in conventional isolation protocols affect the temperature during the perfusion procedure. All these experimental variations are excluded by the use of the standardized gentleMACS Perfuser developed in this study. In addition, primary liver cells are highly sensitive to hypoxia, mechanical stress and other factors that impact cell viability. The usage of the disposable C Tube in which the cells are released from the perfused liver by slow and soft stirring in the gentleMACS Octo Dissociator with Heaters reduces the mechanical stress that occurs in the conventional isolation procedure when stretching and tearing the tissue with tweezers for tissue disaggregation.

Contaminants are other factors that interfere with the reproducibility of experiments. The conventional isolation method proposed by Seglen requires a technically demanding in situ circulating perfusion step prone to contamination, while the combination of the two disposables (i.e., Perfuser and C Tube) reduces the risk of unwanted microbial contamination during cell isolation. As such, the device fosters adhering to good aseptic cell-culture techniques, which in turn increases the reproducibility of the results obtained with the respective cells.

The fact that the cells can be isolated from resected tissue makes this technology also interesting for those that will isolate hepatocytes from parts of livers or other sources, including human resected liver specimens. Finally, the automated gentleMACS Perfusion Technology offers the possibility to perform up to eight individual perfusion processes in parallel, which might be interesting in the case that cells from livers of animals differing in genotype, gender or age are simultaneously isolated.

Altogether, there are marked differences between standard perfusion and the gentleMACS Perfusion Technology that makes the new device attractive for many laboratories working with primary liver cells (Table 2).

## 4. Conclusions

In sum, the novel device developed here may have the potential to become a new standard technique for isolating all kinds of liver cells from almost all sources. It reduces the variability of primary liver cells, thereby increasing the overall reproducibility of experiments conducted by different persons or laboratories. Combined with preformatted buffers and enzyme solutions, it allows the establishment of standardized protocols (SOPs) which will allow researchers to isolate hepatocytes and other hepatic cell types from different sources in a more consistent way.

## Figures and Tables

**Figure 1 biomedicines-10-02198-f001:**
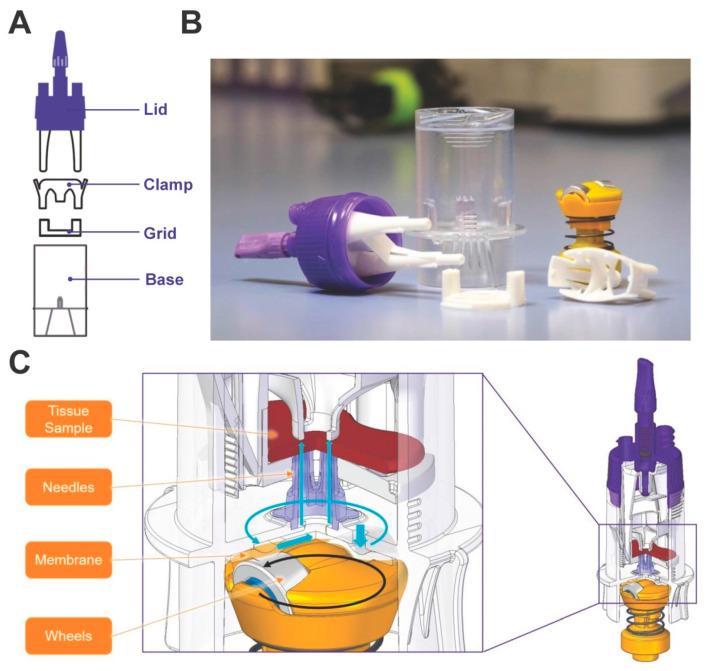
The gentleMACS Perfusion Technology. (**A**–**C**) The Perfuser can be disassembled into four parts, i.e., the Lid, Clamp, Grid, and Base. The Base contains an in-built peristaltic pump which delivers liquids via a needle array into the penetrated liver sample. To aspirate buffer from the inside of the Perfuser, a Pasteur pipette with an elongated tip is introduced through one of the Luer-lock connectors to the Base bottom. The Adjuster on top of the Lid can be used to adapt the penetration depth of the needle array into the liver tissue.

**Figure 2 biomedicines-10-02198-f002:**
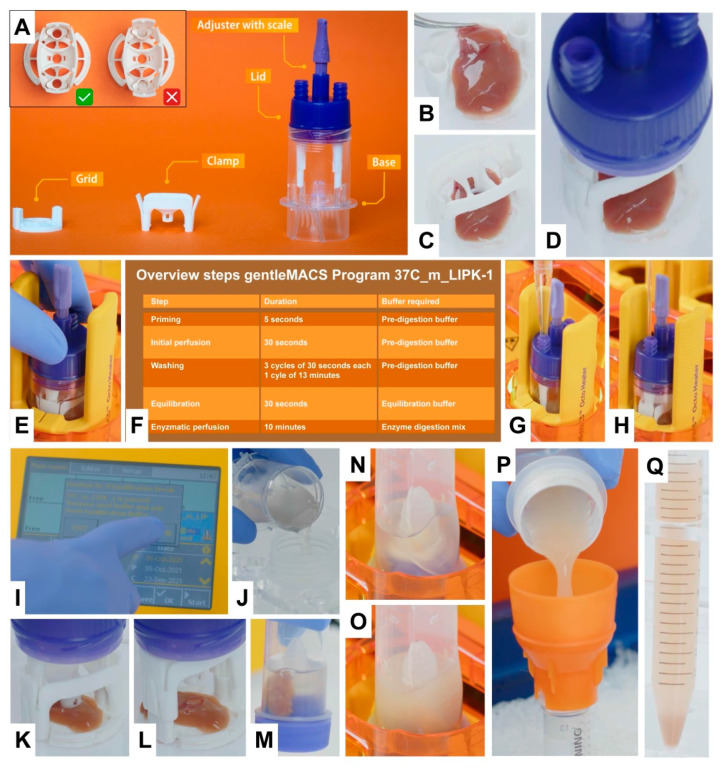
Selected working steps for the isolation of primary mouse hepatocytes using the gentleMACS Perfusion Technology. (**A**) The gentleMACS Perfuser components. The inlet shows how the Clamp should be inserted into the Grid. (**B**,**C**) Prior to perfusion, the liver lobe is placed on the Grid and fixed with the Clamp. (**D**) The Lid connected via the Grid holder to the Grid-Clamp assembly. (**E**) Lid-Clamp-Grid structure screwed into the Base of the gentleMACS Perfuser. (**F**) Overview of program steps of the gentleMACS Program 37C_m_LIPK_1. (**G**,**H**) Addition and removal of buffers after each cycle can be done via one of the Luer locks on top of the Lid. (**I**) Display of the gentleMACS Octo Dissociator with Heaters. (**J**) Filling of the used enzyme digestion mix from the Base into a gentleMACS C Tube. (**K**,**L**) Lifting of the Clamp after completion of the perfusion to enable transfer of the perfused lobe into the gentleMACS C Tube. (**M**) Tissue after transfer from the Grid into the C Tube containing the used enzyme digestion mix. (**N**) gentleMACS C Tube containing perfused liver placed on the gentleMACS Dissociator. (**O**) Cell suspension in C Tube after running Program LIPK_HR_1. The release of hepatocytes from the tissue results in cloudiness of the solution. (**P**) Filtering of cell suspension through the MACS SmartStrainer. (**Q**) Hepatocyte pellet after low-spin centrifugation. The detailed protocol is given in Appendix A.

**Figure 3 biomedicines-10-02198-f003:**
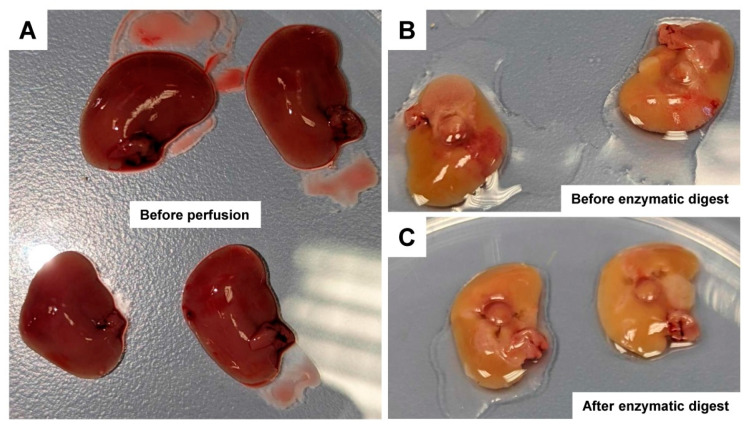
Color changes of the liver tissue during perfusion with the gentleMACS Perfusion Technology. (**A**) Four different liver lobes before perfusion are depicted. (**B**) The upper two liver lobes from (**A**) after perfusion with the pre-digestion buffer. (**C**) The lower two liver lobes from (**A**) after enzymatic digestion.

**Figure 4 biomedicines-10-02198-f004:**
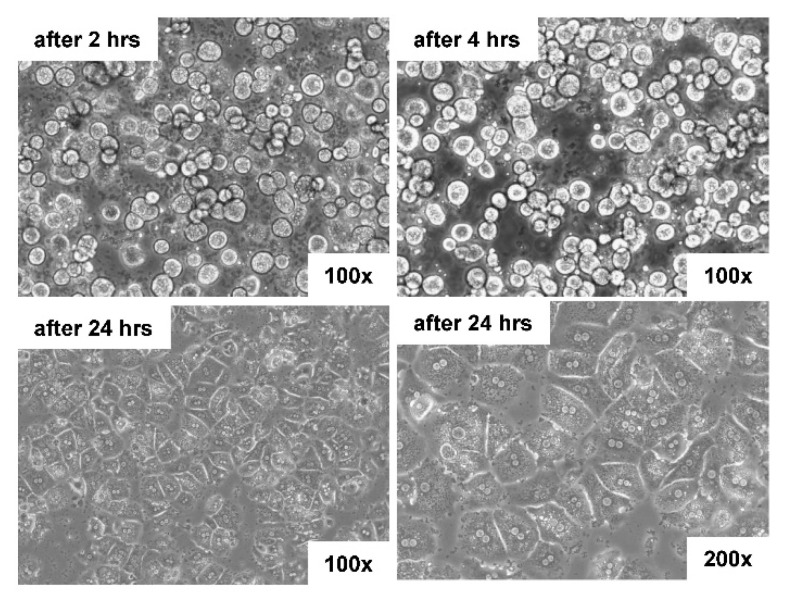
Light microscopic appearance of hepatocytes isolated with the gentleMACS Perfusion Technology. Primary hepatocytes were isolated from a 3-month-old male mouse and seeded in HepatoZYME-SFM medium (#17705021, ThermoFisher, Dreieich, Germany). The medium was changed to Dulbecco’s Modified Eagle’s Medium containing 10% fetal bovine serum, 2 mM L-glutamine, 100 IU/mL penicillin, and 100 µg/mL streptomycin (all from ThermoFisher) two hours after initial plating. Pictures were taken at indicated times and magnifications.

**Figure 5 biomedicines-10-02198-f005:**
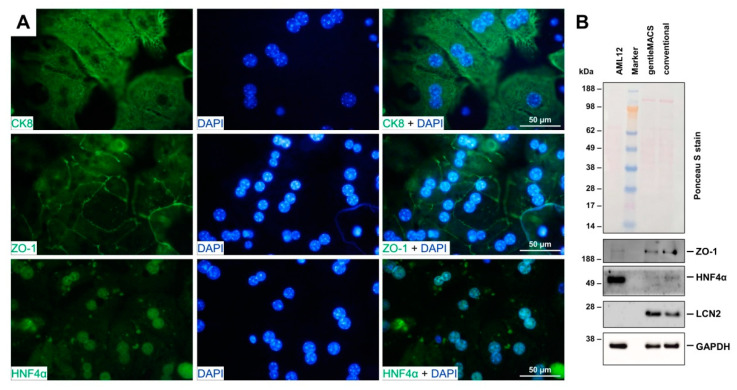
Marker protein expression in hepatocytes. (**A**) Primary hepatocytes were isolated with the gentleMACS Perfusion Technology and subjected to fluorescence immunostaining using rabbit monoclonal anti-cytokeratin 8 (CK8) (#ab53280, Abcam, Berlin, Germany), polyclonal rabbit anti-Zonula occludens-1 (ZO-1) (#85047, Novus Biologicals, Wiesbaden-Nordenstadt, Germany), and polyclonal goat hepatocyte nuclear factor 4α (HNF4α) (#sc-6556, Santa Cruz Biotech., Santa Cruz, CA, USA) for staining. Nuclei were counterstained with 4′,6-diamidino-2-phenylindole (DAPI, Sigma-Aldrich, Taufkirchen, Germany) and images were taken at 1000× magnification using a Leica DMRB fluorescence microscope (Leica, Wetzlar, Germany) equipped with a Nikon Coolpix 5400 camera (Nikon Germany, Düsseldorf, Germany). Brightness and contrast were adjusted for clarity without changing the content. (**B**) Protein extracts from hepatocytes isolated with the gentleMACS Perfusion Technology or conventional isolation procedure were tested using Western blot analysis for expression of ZO-1 (#40-2300, ThermoFisher), HNF4α (#sc-6556, Santa Cruz Biotech.), Lipocalin 2 (LCN2) (#AF3508, R&D Systems, Wiesbaden, Germany) and Glyceraldehyde-3-phosphate dehydrogenase (GAPDH) (#sc-32233, Santa Cruz). The mouse hepatocyte cell line AML12 was taken as a positive control for expression of ZO-1 and HNF4α.

**Figure 6 biomedicines-10-02198-f006:**
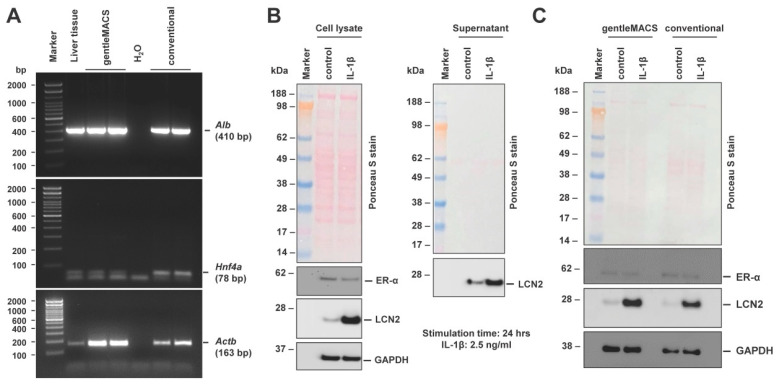
Biochemical features of hepatocytes isolated with the gentleMACS Perfusion Technology. (**A**) Comparative analysis of Albumin (*Alb*) and hepatocyte nuclear factor 4α (*Hnf4a*) mRNA expression in hepatocytes isolated either by the gentleMACS Perfusion Technology or conventional isolation procedure. The RNA was subjected to a standard RT-PCR using the following primers: *Alb-for:* 5′-ggt ctc atc tgt ccg tca gag-3′, *Alb-rev*: 5′-gga aga cat cct tgg cct cag-3′, *Hnf4a-for:* 5′-cca aga ggt cca tgg tgt tt-3′, *Hnf4a-rev*: 5′-ccg agg gac gat gta gtc at-3′, *Actb-for*: 5′-ctc tag act tcg agc agg aga tgg-3′, and *Actb-rev*: 5′-atg cca cag gat tcc ata ccc aag a-3′. Cycle conditions were: 5 min initial denaturation at 95 °C, 1 min at 95 °C, 1 min annealing at 60 °C (*Actb*, 25 cycles; *Hnf4a*, 35 cycles) or 64 °C (*Alb*, 30 cycles), 3 min extension at 72 °C, and final elongation at 72 °C for 10 min. Amplicons were separated in 1.6% (*Alb*) or 2% (*Actb, Hnf4a*) 1 × TBE-buffered agarose gels containing ethidium bromide and visualized using gel imager Gel iX20 (Intas Science Imaging Instruments GmbH, Göttingen, Germany). (**B**) Western blot analysis of protein extracts and supernatants isolated from cultured hepatocytes isolated with the gentleMACS Perfusion Technology that were stimulated with recombinant human IL-1β (#130-093-895, Miltenyi Biotec) or left untreated (control). (**C**) Comparative analysis of IL-1β sensitivity of hepatocytes isolated with the gentleMACS Perfusion Technology or the conventional isolation protocol. Antibodies used were directed against estrogen receptor-α (ER-α) (#MA1-27107, ThermoFisher), Lipocalin 2 (LCN2) (#AF3508, R&D Systems), or Glyceraldehyde-3-phosphate dehydrogenase (GAPDH) (#sc-32233, Santa Cruz Biotech.).

**Table 1 biomedicines-10-02198-t001:** Representative hepatocyte isolations from female mice (n = 33) *.

Number of Animals Used	Strain	Age (Weeks)	Total Hepatocyte Yield **	Hepatocyte Yield/g **	Viability (%) ***
n = 2	CD1	5	7.0 × 10^6^	1.5 × 10^7^	81
			1.9 × 10^7^	5.8 × 10^7^	91
n = 2	BALB/c	9	1.5 × 10^7^	4.5 × 10^7^	85
			5.5 × 10^6^	1.7 × 10^7^	87
n = 2	BALB/c	8	1.1 × 10^7^	3.9 × 10^7^	90
			6.7 × 10^6^	1.9 × 10^7^	88
n = 2	C57BL/6	12	1.4 × 10^7^	2.9 × 10^7^	88
			9.6 × 10^6^	2.0 × 10^7^	88
n = 2	BALB/c	10	1.2 × 10^7^	3.0 × 10^7^	93
			1.2 × 10^7^	3.5 × 10^7^	90
n = 4	BALB/c	10	9.5 × 10^6^	3.2 × 10^7^	93
			3.2 × 10^6^	1.1 × 10^7^	86
			1.1 × 10^7^	3.8 × 10^7^	93
			1.5 × 10^7^	4.8 × 10^7^	92
n = 4	C57BL/6	9	1.1 × 10^7^	3.5 × 10^7^	91
			6.2 × 10^6^	1.7 × 10^7^	85
			1.4 × 10^7^	4.1 × 10^7^	91
			9.8 × 10^6^	2.6 × 10^7^	95
n = 8	BALB/c	8	1.2 × 10^7^	3.4 × 10^7^	91
			1.0 × 10^7^	2.5 × 10^7^	91
			1.6 × 10^7^	5.4 × 10^7^	90
			1.1 × 10^7^	3.3 × 10^7^	93
			7.8 × 10^6^	2.4 × 10^7^	90
			1.3 × 10^7^	4.1 × 10^7^	85
			1.5 × 10^7^	4.3 × 10^7^	90
			7.2 × 10^6^	1.9 × 10^7^	84
n = 3	BALB/c	6	9.3 × 10^6^	2.3 × 10^7^	84
			1.1 × 10^7^	3.0 × 10^7^	83
			7.4 × 10^6^	2.2 × 10^7^	86
n = 4	C57BL/6	9	1.4 × 10^7^	4.6 × 10^7^	90
			1.7 × 10^7^	5.2 × 10^7^	94
			1.0 × 10^7^	5.9 × 10^7^	90
			1.7 × 10^7^	5.6 × 10^7^	88

Notes: * For this set of experiments, hepatocyte isolation was exclusively done from the left lateral lobe; ** Yield is given either as total yield or alternatively as yield per g liver tissue; *** For determination of cell viability, the cells were stained with propidium iodide (final concentration: 1 µg/mL) and directly analyzed by flow cytometry. Cells with high forward and side scatter were selected, gated on simultaneous autofluorescence in the UV and FITC channels representing typical features of hepatocytes and evaluated for the existence of propidium iodide intensity.

**Table 2 biomedicines-10-02198-t002:** Differences between the standard perfusion and the gentleMACS Perfusion Technology.

Feature	Conventional In Vivo Perfusion	gentleMACS Perfusion Technology
Approach	in situ	ex vivo
Perfusion environment	anaesthetized animal	resected tissue
Animals	best working with adult animals (cannulation of veins is necessary)	flexible in regard to age of animals (no cannulation of veins necessary)
Animal ethics approval	necessary	not needed
Sample material flexibility	no (whole organ)	yes (lobe rather than whole organ is recommended)
Perfusion enzymes	not standardized (lot-to-lot variations, activities must be determined by customer)	standardized (consistent enzyme activities contained in the corresponding Liver Perfusion Kit)
(Difficult) manual cannulation of liver vessels needed	yes	not needed
Sample processed in a closed system	no	yes
Perfusion steps and volumes (mL)	highly variable	7 × 8
Collection of used perfusion liquid	no	yes
Other lobes of the same liver available for other experiments	no (at least even more difficult to process)	yes
Robustness of needle insertion into the tissue	low-high (depending on user skills)	high (automatically done by the consumable)
Flow rate during perfusion	to be set by user	predefined by the device
Sterility of equipment	no (sterilization needs preparation time)	yes (usage of disposable items)
Sample size limitation	no (open system)	yes (closed system)
Needed instrumentation	peristaltic pump with tubing	gentleMACS Octo Dissociator with Heaters and gentleMACS Perfusion Sleeve
Needed one-way consumables	syringe needle, buffer reservoirs	gentleMACS Perfuser (for perfusion) and gentleMACS C Tube (for dissociation of perfused liver)
Ease of parallelization of perfusion process	low (would require additional peristaltic pumps and complex nesting of work steps)	high (Up to 8 cell preparations can be performed simultaneously on one instrument)
Disruption of perfused liver tissue	manually (mincing of tissue with forceps)	automated (within C Tube driven by instrument)

## Data Availability

The data that support the findings of this study are available from the corresponding author upon reasonable request.

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
