# Peer review of "Isolation of Hepatocytes from Liver Tissue by a Novel, Semi-Automated Perfusion Technology"

_biomedicines, 2022, doi:10.3390/biomedicines10092198_

Round 1
Reviewer 1 Report
In the short communication article “Isolation of hepatocytes from liver tissue by a novel, semi-automated perfusion technology”, Poggel et al present a novel collagenase perfusion method for hepatocyte isolation that is performed in vitro. This method uses a purchased single-use commercial component (provided by Militenyi Biotec) as well as the accompanying GentleMacs Dissociator with heater. It claims to produce hepatocytes that are comparable to the widely used standard in vivo collagenase perfusion protocol. The data presented demonstrates that they can isolate hepatocytes using this novel system. The authors do not include the entire methodology used (what solutions are they using to perfuse, how do they isolate the liver tissue, which lobe is used, how is the tissue positioned in the chamber, what is the catalog number of the apparatus that they are using?) nor do they compare the hepatocytes isolated using this methodology to those isolated using the traditional in vivo perfusion methods. The paper is easy to read and otherwise well written. Standardizing the collagenase perfusion method of isolating hepatocytes from rodent liver using a relatively easy to use/disposable device is an advance in the field and could allow more researchers to isolate primary hepatocytes and thus such work could make primary hepatocyte isolation a more accessible task, however the data and methodology in their article does not support their claims that such hepatocytes are similar to those isolated using traditional techniques nor do they provide enough detail in their methods to reproduce their experiments.
Major Issues:
1) The authors should provide a detailed isolation protocol.
2) The authors must directly compare the hepatocytes isolated using the Perfuser to those isolated using the original in vivo collagenase perfusion method- essentially demonstrating that the hepatocyte data presented in Figure 2 is similar (or better, more efficient etc) to those isolated using the traditional collagenase perfusion .
3) The IF panel would be more convincing if the nuclear protein HNF4 (a definitive hepatocyte marker) and DAPI were separated.
Minor issues
1) The authors state that an advance of the Perfuser is that it can be used on younger animals with smaller veins but the authors use 3 month old male mice for all of their analysis.
2) Please address why only male mice were used in these studies.
Author Response
Dear reviewer 1,
many thanks for reviewing our paper. Please find our comments to your suggestions in the attached pdf-file.

Reviewer 2 Report
This manuscript applied a novel perfusion system to isolate primary mouse hepatocytes. This new perfusion method will greatly reduce the suffering of animals and failures in the cannulation of the vena cava or portal vein while maintaining a reasonable yield of viable cells. However, this manuscript needs a material and method section to specify animal models used in this protocol (i.e., age, liver weight, etc.), describe perfusion procedures in detail (i.e. perfusion conditions including speed of perfusion flow and time length), and describe the liver dissection procedures and density-based centrifugation conditions in detail.
Coagulation of blood in the liver is an issue for conventional primary mouse hepatocyte protocol, which elongates the perfusion time and inhibits subsequent collagen digestion. Please present photos of mouse livers before perfusion, after calcium-free buffer perfusion, and after collagenase digestion. Also, the author needs a table or graph to present the yield and viability of hepatocytes from each experiment.
In line 88, the author presented a typical cell yield isolated from "one liver lobule". The hepatic lobule is the anatomic unit of the liver. Please clarify if the cell yield is from one liver lobe or a single mouse liver.
I recommend the author list differences between the conventional perfusion protocol and the new protocol in a table.
Author Response
Dear reviewer 2,
many thanks for reviewing our paper. Please find our comments to your suggestions in the attached pdf-file.

Round 2
Reviewer 1 Report
The authors have mainly addressed my previous concerns.
Minor:
1) It is difficult to read Fig 2F.
Reviewer 2 Report
Dear author:
Thank you for answering my questions and concerns. The data presentation and images of processed livers are excellent now.